# Emerging Canonical and Non-Canonical Roles of Granzyme B in Health and Disease

**DOI:** 10.3390/cancers14061436

**Published:** 2022-03-10

**Authors:** Ellis Tibbs, Xuefang Cao

**Affiliations:** 1Department of Microbiology and Immunology, School of Medicine, University of Maryland Baltimore, Baltimore, MD 21201, USA; ellis.tibbs@som.umaryland.edu; 2Marlene and Stewart Greenebaum Comprehensive Cancer Center, University of Maryland Baltimore, Baltimore, MD 21201, USA

**Keywords:** Granzyme B, cancer, cytotoxicity, immune response, extracellular matrix remodeling, tissue development

## Abstract

**Simple Summary:**

Granzyme B (GzmB) is a potent cytotoxic molecule that is used by cytotoxic T lymphocytes and natural killer cells to kill targeted infected or cancerous cells. Since its discovery, there has been growing evidence showing more roles besides cell death such as tissue development and repair. Recent developments in the study of GzmB have given scientists the opportunity to bridge the concepts of cytotoxicity and development, showing that its expression, secretion and function are more intricate than originally suggested. In this review, we provide an overview of the emerging knowledge of GzmB, both canonical and non-canonical, and how these features can be used to elucidate our understanding of GzmB under different clinical conditions.

**Abstract:**

The Granzyme (Gzm) family has classically been recognized as a cytotoxic tool utilized by cytotoxic T lymphocytes (CTL) and natural killer (NK) cells to illicit cell death to infected and cancerous cells. Their importance is established based on evidence showing that deficiencies in these cell death executors result in defective immune responses. Recent findings have shown the importance of Granzyme B (GzmB) in regulatory immune cells, which may contribute to tumor growth and immune evasion during cancer development. Other studies have shown that members of the Gzm family are important for biological processes such as extracellular matrix remodeling, angiogenesis and organized vascular degradation. With this growing body of evidence, it is becoming more important to understand the broader function of Gzm’s rather than a specific executor of cell death, and we should be aware of the many alternative roles that Gzm’s play in physiological and pathological conditions. Therefore, we review the classical as well as novel non-canonical functions of GzmB and discuss approaches to utilize these new findings to address current gaps in our understanding of the immune system and tissue development.

## 1. Introduction

Since the initial discovery of Granzymes (Gzm’s), they were believed to be the executor of cell death produced by cytotoxic T lymphocytes (CTL) and natural killer (NK) cells to target infected cells and transformed tumor cells. Through the granule exocytosis pathway, with the additional help of perforin creating a pore, Gzm’s traffic into the cytosol of the target cell and cleave critical substrates, ultimately leading to apoptosis of that cell [1]. Granzyme B (GzmB) in particular targets BH3-interacting domain death agonist (Bid) and inhibitor of caspase activated DNase (ICAD) in the cytoplasm but can also translocate into the nucleus, targeting lamin B and PARP1 [2,3,4,5,6]. Granzyme A (GzmA), the other major granzyme, was thought to not play a primary role in cytotoxicity, but recent findings show a function in pyroptosis, a proinflammatory form of cell death [7]. GzmA can also traffic into the nucleus and the mitochondrial matrix to induce damage [8,9,10,11]. Many members of the Gzm family, including GzmB and GzmA, have been shown to regulate extracellular matrix remodeling, angiogenesis and organized vascular degradation, all important processes in tissue repair and organ development. Do we truly understand the multivariate story of GzmB? To address this question, we review the classical and non-classical roles of GzmB outside of their cytotoxic function, as well as novel approaches to utilize these new findings to address current gaps in our understanding of the immune system and tissue development.

## 2. Canonical Roles of Granzyme B

Cytotoxic T lymphocytes (CTL) and natural killer (NK) cells utilize Granzyme B (GzmB) and Perforin (Prf-1) molecules in cytotoxic granules to kill virally infected cells and tumor cells. In a classical scenario, antigen-presenting cells (APCs) present a pathogenic antigen on its major histocompatibility complex (MHC) along with costimulatory ligands to CTLs. This interaction gives the CTLs the signal to migrate through the system, surveying the expression of this pathogenic antigen found on the MHC [12]. Once the CTL has come into contact with a cell expressing this antigen, it will begin the process of killing through the creation of an immunological synapse [13]. Unlike CTLs, NK cells do not require antigen-specific recognition to kill target cells. NK cells have the ability to illicit cytotoxicity against MHC class I-deficient target cells, which is a response to viral infections and tumor transformation [14,15].

In order for CTL and NK cells to induce their cytotoxic function utilizing GzmB, they must form an immunological synapse and polarize their cytotoxic granules to the site of membrane contact with the target cell [16,17] (Figure 1). The plasma membranes between the target cell and the killer cell form an area of tight contact in which an adhesion ring is created. This adhesion ring allows for granules to polarize towards this location. The granules are secreted at a site separate from but in proximity to the signaling portion of the immunological synapse, likely to decrease the possibility that the cytotoxic molecules disrupt the synaptic interaction [13]. These observations suggest that the direction of secretion is dictated by the immunological synapse structure or, at minimum, extracellular interaction to polarize granules to allow for a specific release to the target cell [18].

As we study the importance of GzmB in the immune response, it is critical to acknowledge the redundancy in the production mechanisms of these cytotoxic molecules. For CTLs and NK cells, proinflammatory cytokines or activating chemicals correlate with GzmB production. NK cells can generate GzmB by binding Type I IFN or IL-18, or an increase in the interaction with the activating ligand, as well as through antibody-dependent cell-mediated activation using their Fcγ receptor (CD16) [19,20,21]. CTLs utilize their TCR–MHC interaction along with the co-stimulation molecule and CD3 and CD28 stimulation [22,23,24,25]. A chemical approach with phorbol 12-myristate 13-acetate (PMA) and ionomycin, or high-dose IL-2 or IL-15 without an antigen can also activate both NK and CTL cells’ production of GzmB and Perforin [26,27,28,29,30]. Receptors for these two cytokines share a common gamma chain (γ_c_, CD132) and β chain (IL-2Rβ, CD122) [31]. Signaling through either IL-2 or IL-15 receptor is transmitted intracellularly via the JAK1/3-STAT3/5 pathway [32].

### 2.1. Granzyme B Regulation in CTL and NK Cells

Although there is redundancy in the production mechanisms of GzmB, due to its cytotoxic nature, GzmB production and secretion are strongly regulated at both the transcriptional and translational levels. Within T lymphocytes, the first step is the transcriptional activation of GzmB involving activation of the T cell receptor and co-stimulation, along with cytokines [22,23,24,25]. GzmB protein is still held within the activated T cell until the second step, appropriate stimulation of its TCR with the infected target cell. Although they use different signaling mechanisms to illicit killing, NK cells must undergo a similar series of steps to release GzmB into the extracellular environment [33,34].

### 2.2. Regulating Self-Inflicted Damage

As one can imagine, equipping and storing such a cytotoxic molecule is a risk to the cell secreting it. There have been several studies that investigate the production of GzmB potentially causing self-inflicted damage or even self-killing (suicide). One group discovered that in vitro activated human Tregs generate GzmB along with its inhibitor, Serine Protease Inhibitor 9 (Serpin9). However, Serpin9 levels did not fully prevent Tregs from undergoing apoptosis [35]. They also discovered that this accumulation of GzmB intracellularly was due to granule leakage, which induces the cleavage of cytoplasmic and nuclear substrates, indicative of apoptosis pathway activation. Serpin9 inhibits GzmB by an irreversible suicide substrate mechanism, locking the protease with the inhibitor [36] (Figure 1). When CTLs were deficient of Spi6, a mouse homolog of Serpin9, there was a defect in survival that was GzmB-dependent [37]. These results were also found in Regulatory T cells (Tregs) and invariant Natural Killer T cells [38,39]. In fact, the Spi6-deficient Tregs were rescued from apoptosis with the addition of a GzmB inhibitor [38]. In our attempts to understand these findings, there seems to be an evolutionary shift to not alter the “leakiness” of cytotoxic granules but to create mechanisms that address the consequences from said leakiness.

### 2.3. Transcriptional Regulation

GzmB transcription is regulated by the binding of transcription factors upstream of its promoter region, including the activating transcription factor (ATF) and cyclic AMP–responsive element-binding protein (CREB) interaction, activator protein-1 (AP-1), Ikaros, core-binding factor (CBF/PEBP2), Runx3 and T-box factors (T-bet) [40,41,42,43,44]. These transcription factors help to regulate the expression of GzmB in a synergistic fashion. In fact, a mutation or deficiency in any transcription factor or binding site may diminish the expression of GzmB [43,44,45]. It is important to note that the expression of a few transcription factors for GzmB production in mature cells are also important for the differentiation of CTLs, such as Runx3, T-Bet and Eomesodermin (Eomes) [44]. However, Runx3 has also been found to be critical for the generation of certain myeloid lineages [46,47]. When analyzing the expression of GzmB in non-lymphocytes, it is critical to establish transcription factors that are indicative of its expression. Due to its regulation, determining the coexpression of known or novel transcription factors in non-lymphocytic GzmB-producing cells is a tedious yet necessary task to progress the field to study this cytotoxic molecule.

The post-transcriptional regulation of GzmB is evident in the described cell types, although the mechanisms involved in this regulation are not fully understood. Resting mouse NK cells make a large number of GzmB transcripts but no GzmB protein is expressed or secreted [48]. The reason has not been investigated; however, one reason could be the licensed/unlicensed NK cell paradigm, in which a certain percentage of NK cells can quickly begin killing without the need to be primed [49]. However, once these NK cells have been activated, there is a significant increase in GzmB protein levels, with relatively little change in the transcript levels [48]. In the context of a highly cytotoxic cell type producing a powerful cytotoxic molecule, it would make sense to create a two-step authentication to allow production and release, first priming and then activation. In contrast, human mast cells express high levels of GzmB transcripts and relatively low levels of GzmB protein following stimulation [50]. Regardless of the cell type expressing GzmB, there must be strict regulation of the production and secretion of this cytotoxic molecule to confirm that damage to neighboring cells is necessary.

### 2.4. Translational Regulation

Lastly, the post-translational regulation of GzmB is performed in a similar approach to many other potent proteases. GzmB is synthesized as a propeptide and trafficked to acidic secretory lysosomes through the mannose 6-phosphate receptor pathway [51,52]. At this point, GzmB is then activated by the dipeptidyl peptidase Cathepsin C before degranulation to its target cell [53,54]. Each point in this pathway is dictated by a critical protein or enzyme. If novel cells are producing GzmB, it is important to decipher what form of GzmB is made and if these necessary steps are initiated in the cell types of interest.

## 3. Non-Lymphocytic Granzyme B Production

### 3.1. Mast Cells

Many studies have investigated the production of GzmB by mast cells. Mast cells are long-lived cells derived from hematopoietic stem cells. These cells migrate to their destined tissue while in their immature state, maturing once they are at the determined location. Due to these locations being near surfaces exposed to the environment, mast cells are one of the first types of cells of the hematopoietic-immune system to come into contact with allergens and other antigens [55]. Murine mast cells derived from bone marrow have been shown to produce and release the enzymatically active form of GzmB upon ligation of FcεR1 [56]. Release of GzmB is independent of Prf-1, which induces anoikis, a form of cell detachment. Anoikis has been shown to play a critical role in Prf-1-independent killing [57]. This production and release of GzmB has been shown to act as a protumoric factor by decreasing the efficacy of vascular endothelial growth factor receptor (VEGFR)-targeted anti-angiogenic therapies [58]. They found that GzmB can induce the release of fibroblast growth factor (FGF-1) and granulocytic-monocytic colony-stimulating factor (GM-CSF) from the extracellular matrix. These data correlate with a study that found that GzmB can release VEGF from the extracellular matrix [59]. One way to illicit the production of GzmB by these mast cells is through antibody-mediated activation. Binding of the antibody to killer-cell Ig-like receptor 2DL4 (KIR2DL4/CD158d) induces GzmB secretion from Laboratory of allergic diseases 2 (LAD2) and peripheral blood (PB) mast cells, while the use of a Src homology 2-containing protein tyrosine phosphatase (SHP-2) inhibitor significantly decreases this phenotype by decreasing the rate of degranulation as well as the rate of GzmB production [60]. 

### 3.2. Myeloid Derived Suppressor Cells

It has been extensively studied how cancers can manipulate the surrounding tissues to promote their survival and progression. Myeloid-Derived Suppressor Cells (MDSCs) are heterogenous innate immune cells that are found in many different pathologies, but primarily cancer [61]. They originate in the bone marrow and are recruited to sites of inflammation. Tumors use this to their advantage to increase the production and recruitment of highly suppressive MDSCs to the tumor microenvironment [62]. Although this cell population has become quite popular in the field of tumor immunology, much remains unclear in their variety of protumoral mechanisms. Recent findings have shown that human MDSCs as well as human CD68^+^ macrophages are able to produce GzmB; however, the mechanism for this production is still to be established [63,64]. There has been one study that found the expression of both GzmB and Prf-1 in murine MDSCs [65]. This group not only confirmed this expression but also found that this GzmB and Prf-1 secretion is essential for tumor growth. Although there no differences were found in their effect on T lymphocytes, MDSCs deficient in GzmB cannot facilitate the migration of tumor cells compared to WT. This, in addition to GzmB produced by mast cells, prompts the speculation that the production and secretion of GzmB causes extracellular remodeling, which aids tumor migration and metastasis. As this cell population continues to be studied, one concept that still needs to be addressed is the signaling mechanisms needed to produce and release GzmB in these cell types. To generate this population of suppressive cells, the cytokines GM-CSF and IL-6 are very important; however, the downstream signaling motifs are not classically associated with GzmB production. In addition, the hypothesis that MDSCs constitutively express GzmB without a “second” signal should be a cause for concern due to the potent effects of this cytotoxic molecule. If there are no additional signals to help regulate the production or secretion of GzmB, then this can lead to detrimental effects in chronic inflammatory diseases such as cancer. There is still controversy in the expression of GzmB between human and mouse MDSCs. This can be equated to the findings of human B cells expressing GzmB, while murine models to confirm it have been unsuccessful [66]. A simple explanation could be that the mechanisms that regulate GzmB expression are not conserved between humans and mice. 

### 3.3. Signaling Mechanisms for Granzyme B Production in Non-Lymphocytic Cells

To fully understand the roles of GzmB, we must look at the enzyme in many contexts. One context would be the discovery of human plasmacytoid dendritic cells (pDC) that express GzmB following IL-3 stimulation [67]. This stimulation through the IL-3 receptor complex is JAK1-STAT3/5-dependent. It has also been found that IL-10 stimulation following IL-3 increases the population of GzmBs producing pDCs, while stimulation of TLR7/9 and CD40 decreases this effect. As IL-10 utilizes the JAK1-STAT3/5 pathway, it does not seem to be redundant, as IL-3 prior to stimulation is required to enhance GzmB production [67,68].

The results illustrate an interesting idea, namely that CD40 stimulation is normally a pro-inflammatory signal, but under GzmB-producing conditions in pDCs, CD40 stimulation leads to a decrease in GzmB production. Once ligation occurs, there is recruitment and trimeric clustering of the TNF Receptor Associated Factor family of proteins (TRAFs), with TRAF6 being the most important in dendritic cells [69,70]. Normally, TRAF6 recruitment leads to the production of pro-inflammatory cytokines such as IL-12p40 and IL-6 [71]. CD40 can synergize with TLR9 signaling to enhance the production of Type I IFN [72]. JAK1-STAT3/5 signaling in one cell type means the production of cytotoxic molecules, while a similar signaling cascade in another cell type does not. To add another layer of complexity, the classical anti-inflammatory signal IL-3 still utilizes the JAK1-STAT3/5 pathway differently but now leads to the production of GzmB, a classical cytotoxic molecule.

Connecting the story of signaling for GzmB production and MDSCs is rather difficult. Recent findings have shown that GR-1^+^CD11b^+^ MDSCs do in fact express GzmB [65]. However, in the generation and release of GzmB, the MDSCs were not given any additional stimulation other than the culture techniques used to generate the heterogenous cell population: conditioned media from Granulocyte-Macrophage Colony Stimulating Factor (GM-CSF) producing tumor cells. This can also be interpreted as MDSCs being able to express GzmB ubiquitously as long they are in their immature state. This would be the first cell type that does not need a first or second signal to produce GzmB. GM-CSF binding to its receptor allows for the utilization of the JAK2/STAT5 pathway, which is completely different from the aforementioned signaling pathways that allow for GzmB production [73]. There needs to be more investigation of the signaling mechanisms of GzmB by non-lymphocytic cells.

## 4. Non-Canonical Roles of Granzyme B

### 4.1. Role of Granzyme B in Inhibiting Infectious Diseases

One topic that has not been heavily investigated is the ability of GzmB to play antipathogenic roles that are independent of cellular apoptosis. There are historic data of GzmB clearing infections due to the apoptosis of infected cell types; however, GzmB has been found to cleave multiple viral proteins that play a critical role in viral replication and/or host evasion, such as herpes simplex virus (HSV) ICP27, HSV ICP4, varicella-zoster virus (VZV) ORF4, and VZV ORF62 [74]. This is believed to be a secondary mechanism of inhibiting viral spread due to HSV and VZV being able to inhibit both intrinsic and extrinsic apoptotic pathways [75]. GzmB has also been shown to cleave eIF4G3, a host protein that is involved in viral translation, independently of caspase activity [76]. Another mechanism suppressing viral replication was found when GzmB disrupts splicing of newly synthesized mRNAs in a caspase-dependent manner [77].

GzmB, with the help of Prf1 and Granulysin (GNLY), is delivered to intracellular parasites and bacteria inducing the generation of reactive oxygen species (ROS) [78,79]. For parasites, the generation of superoxide anions disrupts the mitochondrial potential, inducing apoptosis. Interestingly, the death of intracellular parasites occurs independently of and before host cell death, a potential way to limit parasite spreading. In the case of bacteria, GzmB proteolytically disrupts electron transport chain complex I and superoxide dismutase (SOD), inflicting irreparable damage and inhibiting any oxidative stress response of aerobic bacteria. In another study, it was found that GzmB disrupts bacterial protein synthesis and metabolism through ribosome disassembly, allowing the infected cell to undergo apoptosis [80]. GzmB has also been found to decrease bacterial virulence through the cleavage of multiple secreted and membrane-exposed virulence factors including *Listeria monocytogenes*, *Salmonella typhimurium*, and *Mycobacteria tuberculosis* [81].

### 4.2. Cytotoxicity towards Other Immune Cells

Besides of the use of GzmB to kill targeted infected or cancerous cells, there have been studies that show GzmB being used as a tool to suppress the immune response, particularly by CD4+CD25+ Regulatory T cells (Tregs). It was found that Tregs suppress B cell proliferation by inducing death via a granzyme-dependent pathway. The results also show that Tregs preferentially kill B cells that present specific antigens rather than through a bystander effect [82]. In another set of experiments, it was found that Tregs inhibit the process of tumor progression by killing leukemic B cells in vitro [83]. This is an interesting finding because the common belief is that Tregs aid cancer, but since this is a cancer of the immune system, Tregs may view them as a chronic immune response and mediate suppression. We can surmise that this is a mechanism that inhibits the induction of autoimmune antibodies. Another example of this suppressive mechanism is the ability of human Tregs to kill immature myeloid dendritic cells and CD14+ monocytes. Following a 4 h coculture, there was significant amount of death of dendritic cells. Interestingly, maturation of myeloid dendritic cells significantly decreased this cell death [84]. Although these results are critical to understanding the roles of Tregs in immunity, one concept that still needs to be addressed is how Tregs receive the signal for cytotoxic granule release. The killing of dendritic cells was not antigen-specific, nor was it established that an immunological synapse was formed to illicit this killing. 

Another interesting finding of GzmB is its role in contracting the clonal proliferation of T cells through the expression of Mannose-6-phosphate receptor (M6PR) (Figure 2). M6PR is upregulated on activated T cells and has been shown to aid in the recruitment of protein kinases towards the immunological synapse [85]. As a versatile receptor, M6PR has been shown to bind to GzmB to uptake it into the cell [86]. M6PR expression correlates with Treg-dependent GzmB-mediated apoptosis in that a high expression led to greater apoptosis and low M6PR expression resulted in a significant decrease in apoptosis in effector T cells [87]. In the early phase of *Listeria monocytogenes* infection, the majority of antigen specific CTLs expressed high levels of M6PR, while CTLs that expressed low levels of M6PR survived the early phase of contraction [87]. Although these findings have been shown to be mediated by Tregs, the mechanism for the release of GzmB into the extracellular environment has not been investigated. Antigen-specific T cells were killed in a GzmB-dependent process; however, the dependence of an immunological synapse for this killing has not been investigated. Further in this review, we discuss potential reasons of Granzyme B release without an immunological synapse.

Understanding the mechanism behind this receptor-mediated uptake of GzmB could help overcome the obstacles in the targeted delivery of GzmB. Over the past decade, there have been promising data supporting specific delivery of GzmB using genetic fusion with an antibody or derivative of natural ligands to extracellular surface receptors as a therapeutic tool to combat cancer by increasing expression of pro-apoptotic genes and decreasing the tumor burden in mice. However, one common theme is the presence of off-target effects [88,89,90,91]. Studying the receptors that bind to GzmB could allow investigators to develop a better combination of a fused GzmB-antibody/ligand with mutations in sites that would not allow receptors of off-target cells to engage.

### 4.3. Cell Type Differentiation

Recent studies have shown that GzmB may play an important role in immunological development. Cells that are developed to become naturally cytotoxic may lead to inherently greater levels of the Spi6/Serpin9 to inhibit possible programmed cell death or even fratricide, the killing of neighboring cells during expansion. This could also be achieved by cells that are not classically cytotoxic. For example, one group found that T-helper 2 (Th2) cells are induced by the help of vasoactive intestinal peptide (VIP). The effect of VIP on these cells prevents the upregulation of GzmB and also promotes a survival effect by decreasing Fas ligand (FasL) and Prf-1 expression [92]. Recent discoveries can add to this concept of immunological development. Hoek et al. found that GzmB skews CD4+ T cells away from Th17 phenotype during differentiation. In vivo activation of GzmB-deficient CD4+ T cells resulted in a significant increase in IL-17 production, graft vs. host disease severity, and an increase in donor-derived T cell reconstitution [93]. These data represent not only a novel role of GzmB in T cell differentiation but also a potential suppressive role. Not having GzmB present during the proliferation and differentiation seems to exacerbate pathogenic responses. Another group discovered that GzmB can have intracellular mechanisms that are Prf-1-independent. In fact, they found that this Prf-1-independent mechanism results in nonlethal DNA damage, which leads to phosphorylation of Interferon (IFN) regulatory factor-3 (IRF-3) [94]. These results could connect with what Hoek et al. published. Early GzmB production during CD4+ T cell differentiation may augment the immune response via nonlethal DNA damage, creating a feedforward loop to create more Th1 cells (Figure 2). This also coincides with data that suggest the expression of GzmB before the expression of Prf-1. [95,96]. There are still questions that should be addressed if this concept does prove to be true. For example, are the cells that develop to be autoinflammatory express different levels of Spi6/Serpin9? Does GzmB expression by T cells play a greater role in the peripheral suppression outside of Tregs?

### 4.4. Granzyme B Expression in Cancer

In this review, we have summarized data that focus on the specific cellular production and regulation of GzmB, some promoting cancer growth. Our classical understanding of GzmB is that it is needed to suppress tumor activity; however, some studies have shown something different. In a study focusing on oral squamous cell carcinoma, there was higher prevalence of GzmB expressing Tregs compared to normal patients [97]. This has been reproduced in many other studies focusing on cancer. Pan-cancer expression analysis showed higher GzmB mRNA expression in many cancers such as cholangiocarcinoma, glioblastoma multiforme, head and neck squamous cell carcinoma, kidney renal clear cell carcinoma and stomach adenocarcinoma compared to normal tissues. However, cancers such as lung adenocarcinoma and lung squamous cell carcinoma show low GzmB expression [98]. What was also interesting was the range of GzmB expression within the same cancer type. This provides an important analysis of GzmB expression. We analyzed pan-cancer expression and found that some cancers such as uveal melanoma and pancreatic adenocarcinoma have a better probability of survival when there is minor GzmB transcription compared to a large amount [99]. This seems to correlate greatly with FoxP3 expression in some patients (Figure 3B). The majority of cancers showed a greater probability of survival with greater expression of GzmB (Figure 3A). What was interesting was the finding that FoxP3 expression also increased the probability of survival in cancers such as head and neck squamous cell carcinoma and stomach adenocarcinoma (Figure 3C). One interesting finding was that some cancers showed a large amount of FoxP3 expression with minor GzmB expression such as lower grade glioma (LGG), invasive breast carcinoma (BRCA) and ovarian serous cystadenocarcinoma (OV). Understanding the correlation with FoxP3 and GzmB under different tumor conditions may prove to be important in the generation of tumor targeting therapies.

### 4.5. Extracellular Matrix Remodeling

The formation of an immunological synapse, as described earlier, has classically been associated with a directed secretion of GzmB. However, there may be more novel mechanisms for GzmB being released into the environment that are worth investigating. 

During the progression of infection and inflammation, there is an increase in extracellular GzmB in circulation [100,101]. This is most likely caused by the constitutive release of GzmB from primed NK and CTL cells without synapse formation at a target cell or a release as a result of binding to extracellular matrix (ECM) proteins [18,52,102]. In fact, one group found GzmB to be critical for lymphocyte migration due to basement membrane remodeling. GzmB allows for CTL transmigration in postcapillary venules and homing to the site of infection [103] (Figure 1). Although GzmB-deficient CTLs were able to penetrate the capillary vessel walls, they were incapable of completely passing through. This would suggest that GzmB facilitates the remodeling of the basement membrane to allow for the rigid nucleus of the lymphocyte to pass through. In fact, this aligns with the concept that the multilobular nuclei of the neutrophil allow for the cell to transmigrate through smaller holes with ease and without the need for membrane remodeling. The release of GzmB into the environment in this manner is separate from the immunological synapse formation but may be similar to synaptic release in which a cleft formation allows for granule polarizing for release [13]. However, one problem that was mentioned but not addressed is that when degranulation was inhibited, the CTLs had impaired transmigration, but to a lesser extent than GzmB deficient CTLs. This suggests that there is yet another mechanism for the release of either enzymatically active GzmB or a proenzyme form that is activated in the environment. Protocols using CD3 and CD28 stimulation result in significant production and release of GzmB into the media. This would suggest that the classical immunological synapse is not required but that the cleft formation could still occur and illicit degranulation.

As research progresses, scientists should use new resources to take a closer look at the past. One example is by utilizing prediction programs to find protease cleavage sites on proteins that are not known to be targeted by GzmB, such as GraBCas and PROSPERous [104,105]. We may view GzmB as a similar protein to Spi6, in that Spi6 will not undergo its function if there is no available target. We must focus on potential targets of GzmB while being attentive to the function after interacting with the targets. With the recent discoveries of Gzm’s playing a role outside of cell death, it is not too farfetched to think it may affect cellular development. Our lab in fact had this idea due to the observation that there is larger lymphocytic colony formation in cells deficient in GzmB compared to WT. Speculating that GzmB may be affecting Notch1 signaling, we analyzed the cleavage sites on the Notch1 receptor and found that there are many potential sites that GzmB can target. Notch undergoes *trans*-activation through binding of one of its ligands, Jagged1/2 or DLL1-4 [106,107,108,109]. To our surprise, we saw a similar trend in cleavage sites of the ligands of Notch1 such as JAGGED and DLL4, as well as other known ligands. In addition, one group using an in vitro system showed that GzmB can affect Notch1 transcriptional activity [110]. If this is to be studied further, it would be important to determine exactly where GzmB acts on Notch1 as it has three domains spanning the membrane and two cleavages that must occur to allow for Notch1 intracellular domain (NICD) nuclear translocation. If GzmB acts on NICD, then this must be a perforin-dependent mechanism, while if GzmB acts on the extracellular domain, then it is perforin-independent. These two different mechanisms could lead to completely different stories that could help shed light on how GzmB affects cellular development.

## 5. Concluding Remarks

Although published evidence for the cytotoxic nature of GzmB is extensive and ever-growing, there is also evidence that supports a greater role of GzmB depending on the context. In the past 10 years, scientists have gained much insight into cells that express GzmB, but there are still unanswered questions on how and why these novel cell types produce and secrete such a cytotoxic molecule. We propose a hypothesis that the expression of GzmB, as well as many other proteins in the Gzm family, are critical for many aspects of development outside of targeted killing of infected or cancerous cells such as inhibition of autoimmunity, transmigration to sites of infection and augmentation of the immune response to generate more GzmB-expressing T helper cells. A better understanding of these non-canonical roles could facilitate important therapeutic approaches in a variety of diseases such as autoimmunity and cancer.

## Figures and Tables

**Figure 1 cancers-14-01436-f001:**
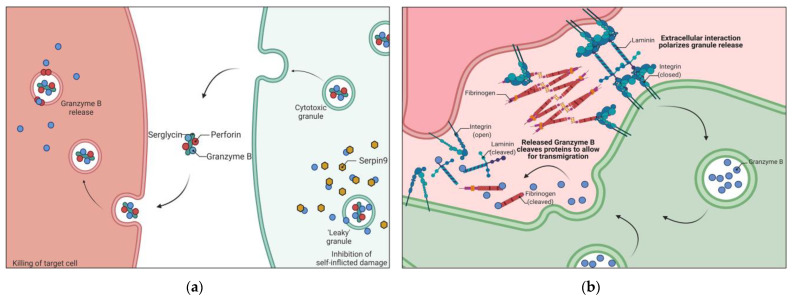
Schematic representation of canonical and non-canonical features of Granzyme B (GzmB). (**a**) Targeted killing of cancer cells by T cells through granule release while inhibiting self-inflicted damage caused by leaky granules. GzmB is introduced into a target cell leading to apoptosis. GzmB leaked into T cells is inhibited by Serpin9. (**b**) Extracellular interaction polarizes granules for release, resulting in cleavage of extracellular factors such as fibrinogen and laminin. These cleavages increase the rate of transmigration of T cells. Serpin9 = Serine Protease Inhibitor 9.

**Figure 2 cancers-14-01436-f002:**
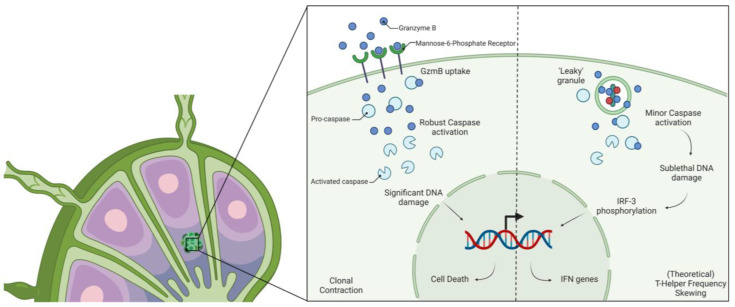
Schematic representation of the novel roles of GzmB in T cell development. (**Left**) Mannose-6 Phosphate receptor expression orchestrates the amount of GzmB introduced into the cell, allowing for robust activation of caspase and clonal contraction. (**Right**) Theoretical mechanism of leaky granule release resulting in the production of IFN genes through minor DNA damage, skewing T helper frequency. IRF-3 = Interferon Regulatory Factor 3; GzmB = Granzyme B; IFN = Interferon.

**Figure 3 cancers-14-01436-f003:**
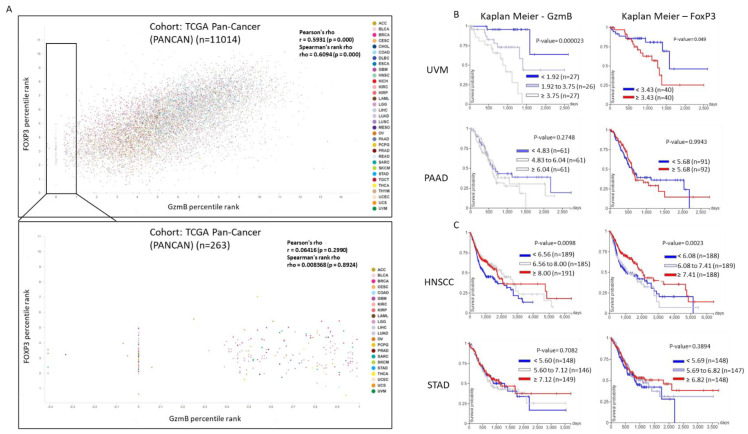
RNAseq analysis in the coexpression of GzmB and FoxP3. (**A**) Relationship between GzmB and FoxP3 expression in the TCGA Pan-Cancer database, identified by UCSC Xena. Color is denoted by cancer type. (**B**) Kaplan–Meier survival plot of GzmB and FoxP3 gene expression extrapolated from available RNAseq data from uveal melanoma or pancreatic adenocarcinoma patients demonstrating higher GzmB expression is associated with worse survival probability. (**C**) Kaplan–Meier survival plot of GzmB and FoxP3 gene expression extrapolated from available RNAseq data from head and neck squamous cell carcinoma or stomach adenocarcinoma patients demonstrating higher FoxP3 and GzmB expression is associated with better survival probability.

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
