# Peer review of "Emerging Canonical and Non-Canonical Roles of Granzyme B in Health and Disease"

_cancers, 2022, doi:10.3390/cancers14061436_

Round 1

Reviewer 1 Report

The authors refocused their original article more on GrzmB, and added two additional chapters to their manuscript addressing the non-canonical roles of GrzmB as well as its role in cancer, respectively. While limited language editing would further improve the presented work, the authors managed to address all minor concerns satisfactorily.

Some minor editorial issues were noted:

  1. ….in defective immune response. (line 20) should read …..in defective immune responses.
  2. …death, so as not to misunderstand…. (line 26) could possibly read ….death, we should be aware the many alternative roles Grzms play…..
  3. Please spell out HSV and VZV upon first appearance (line 254).

Reviewer 2 Report

The Authors have added new parts (or significantly modified it according to the comments done by the Reviewer) to the text, such as parts 4.1 and 4.4, as well as revised several phrases/sentences and words in the text. New references were added to the text.   The comments made by the Reviewer regarding the first version of this work are not critical. Thus, the text can be accepted for publication. 
